# Teaching 21st Century Skills in Saudi Arabia with Attention to Elementary Science Reading Habits

Wadha H. Alotaibi [1] and Amani Khalaf H. Alghamdi [2,*]

1 Curriculum and Pedagogy, Princess Nourah Bint Abdulrahman University, Riyadh 14921, Saudi Arabia; whalotaibi@pnu.edu.sa
2 Curriculum and Pedagogy, Imam Abdulrahman Bin Faisal University, Dammam 31441, Saudi Arabia
* Correspondence: akhalghamdi@iau.edu.sa

**Abstract:** Saudi Arabia's Vision 2030 urges the teaching of 21st century skills. This study involved female elementary science teachers ($n = 55$) and students ($n = 232$) (intentionally taught using science reading and not intentionally taught this way) in two offices of a Saudi Eastern Province school board. Descriptive statistical analyses of teachers' beliefs and students' thoughts on science reading revealed key results. Most teachers believed that students had a high degree of eagerness and passion to learn science, were always passionate and eager to read scientific texts, had a high desire to participate in class discussions of scientific readings, and had a moderate understanding of what they had read. However, the students were more ambivalent. Students who were intentionally taught using science reading scored higher on reading tendencies than those not intentionally taught this way. These statistically insignificant results were interpreted to further garner insights into the Saudi context that is fully grounded in Islam, which deeply values reading.

**Keywords:** 21st century skills; female teachers and primary students; Saudi Arabia; science reading for academics; science reading tendencies

## 1. Introduction

The focus on 21st century skills originated in a 2002 partnership among the United States (US) Department of Education, the National Education Association (NEA), and many American companies and individuals. It now includes many US governments, and the goal is to promote students' acquisition of these skills through the collaboration of schools, businesses, societies, and governments [1,2]. The "21st century skills" include critical thinking, creativity, collaboration, communication, several types of literacy (information, media, technology, digital), flexibility, leadership, initiative, productivity, social skills, and local and global connections [3–5].

To ensure that students learn those skills, several American and global organizations, such as the American Association of Colleges and Universities [6], the Organization for Economic Cooperation and Development (OECD), and the Partnership for 21st Century Learning (P21), have devised the concept of learning in the 21st century. This concept includes student-centered learning strategies that focus on the four Cs: communication, collaboration, critical thinking, and creative thinking [7]. Mastering these high-level thinking skills ensures that the human resources' component of the labor market can respond to changes and developments in the current economy and better compete globally [8].

The US National Science Teaching Association [9] presented a set of recommendations to achieve quality science education that supports 21st century skills. They include (1) integrating these skills into future science teacher preparation and training programs and existing science educational programs and practices as well as (2) into science curricula and (3) ensuring that science evaluation and assessment reflect 21st century skills.

The government of the Kingdom of Saudi Arabia (KSA) is also paying attention to the importance of empowering all learners with 21st century skills. It has included

this imperative in its recent national development plan, Vision 2030: "We will continue to invest in education and training and provide our children with the knowledge and skills needed for future jobs" ([10], p. 71). In addition, the KSA's Education and Training Evaluation Commission (ETEC, formed in 2017) recently submitted a proposal to include 21st century skills in the KSA's public education curriculum, which is a pioneering step. The ETEC [11] suggests that all educational curricula in public education focus on cross-cutting dimensions, which is a new term in the general knowledge and skills section.

The ETEC (2018b) also established the International Conference on Education Evaluation, with its first conference focused on future skills: development and evaluation. One objective was to identify the most important skills that would contribute to both increasing competitive opportunities in employment and achieving professional excellence. The conference was intended to "help educate people about the skills and qualifications highly in demand in today's labor market" ([12], para. 4) and "help design the general framework of future skills" ([12], para. 1). Participants were engaged with the idea of developing the human capital of future Saudi workers per Vision 2030 so that education can add value to their existing skills [12]. Conference recommendations confirmed the urgent need to empower the Saudi workforce with 21st century skills to achieve compatibility between the outputs of educational institutions and the requirements of the new century. Teachers were urged to develop these skills among Saudi learners [12,13]

Recent research on the role of science curricula and science teachers in developing these skills pertains to the ETEC (2018) recommendations. After assessing the content of the KSA's intermediate-level science textbooks, Al-Shahrani and Al Mahfouz [14] reported that the presence of 21st century skills ranged between weak and absent. Abu Rayyah and Al-Shami [15] reported similar results with high school biology textbooks.

Pedagogically, Saudi science teachers often opt for teacher-oriented lectures and lab experiments [16,17]. Middle stage (junior high) Saudi science teachers had a moderate acquisition of 21st century skills prompting the recommendation of preparing and training them in these skills [18,19]. Conversely, Al-Harbi and Al-Jabr [20] concluded that Saudi elementary science teachers had a high level of awareness of 21st century skills. This is advantageous because elementary teachers play a central role in teaching the skill of reading and comprehension with this skill considered one of the 4Cs [7]. Barton and Jordan [21] reported that elementary science teachers were ambivalent about asking students to read scientific texts. One in three used them as reference material only, despite agreeing that "students need to be able to read science" (p. iii).

Osborne [22] added that even when elementary students received a good start on reading and literary work, they often lacked skills in reading science texts. Learning to read science is a challenging task, and elementary "students will do best when their science teachers regard teaching science [reading] as part of their job" (para. 4). Not requiring elementary students to read science is unfortunate because reading such texts requires and teaches critical thinking, analysis, and communication skills and helps students build an in-depth understanding of scientific concepts and principles [21].

It is imperative that students develop an early habit of reading scientific texts so that they have an early start in the acquisition of 21st century skills. They must be able to read science texts while at school and in other aspects of their lives. Reading science leads to academic success, a love for science, and a deeper appreciation of how science is prevalent in daily life (i.e., inventions, innovations, career paths, media, literature) [21]. Reading science texts in the classroom (academic science) can be a precursor to reading science texts in daily life to make sense of science in the world (life science) [23].

Although most Saudi students are accustomed to reading in their social sciences and language courses, the science classes are teacher-centered and involve students listening to and receiving information. Teachers control what is read by preventing students from an in-depth reading of science textbooks. Any lack of interest in or understanding of the scientific content is captured in test results. It is common knowledge that Saudi grade 4 students score low on science in international testing initiatives [24].

Researchers have also observed that Saudi students who are taught science using reading (student centered) are independent, self-motivated learners who search for and select among different scientific texts. These texts tend to show life applications, inventions, and innovations that are closely related to scientific content. This praxis is of interest to students and raises their scientific reading orientation. They practice reading on their own and persist in doing so regardless of the difficulties they encounter. Faced with concepts that are unclear, they push their thinking and search for other explanatory sources. Unconventional reading activities in science raise their passion toward this distinctive type of reading.

## 2. Literature Review

The explosion of knowledge, the rapid development of technology, and the entrenchment of corporate-led globalization are worrying trends for many practitioners, educators, and international organizations with concerns about which skills students should possess [25]. Students need a set of basic skills to succeed in the 21st century, which means that they must have opportunities to develop them through pertinent educational experiences. Discussions on developing 21st century skills through the educational process began in the 20th century and are ongoing. These discussions are supported by international organizations (e.g., the OECD [Paris, France] and World Bank [Washington, DC, USA]) that have an impact on national educational policy [26].

Discussions concerning what these skills are and how they can be developed while students are at school are also underway at many international conferences and strategic sessions, especially with business representatives. Recently, these discussions have begun to focus on implementing 21st century skills in public school curricula and the importance of students' access to this new education [26].

Several studies concerning the most appropriate pedagogy and instructional strategies to impart 21st century skills with a student-centered approach have become pronounced (e.g., flipped classrooms, cooperative problem solving, and project-based learning). Taha [27] affirmed the value of using the science, technology, engineering, and mathematics (STEM) approach to develop 21st century skills in a high school physics unit. Khalil et al. [28] affirmed the value of a flipped classroom to develop these skills in high school biology. Al-Khamisi [29] confirmed that a cooperative problem-solving strategy is useful for developing 21st century skills in middle school students. According to Gkemisi et al. [30], the Jigsaw strategy and problem-based learning were effective in improving cooperation and communication skills. Bell [31] supported project-based learning to foster communication, problem solving, and collaboration.

### 2.1. Reading, Communication, and Reading Habits

Communication, a 21st century skill, entails imparting or exchanging information, meaning, or both. It requires clearly expressing one's thinking and ideas effectively using oral, written, and nonverbal communication techniques in various contexts. To this end, reading is a key communication skill because it is a means of decoding and understanding written text and acquiring new information, ideas, perceptions, and meanings. The ability to read and comprehend what has been read is the crux of communication. Reading can "help provide and facilitate a communication experience that provides the opportunity for authentic and deeper learning." ([32], para. 5)

In the Islamic context, the value of reading becomes evident in the first verse revealed to our Prophet Muhammad (may peace be upon him), in which the Almighty said:

> "Recite in the name of your Lord who (1) Created man from a clot of congealed blood (2) Recite, and your Lord is the most Generous (3) Who taught by the pen (4) Taught man that which he knew not (5)."

Reading and writing in Islam are paramount. "It is an accepted fact that a major priority established by the Qur'an is reading and writing" ([33], para. 5). The first command of the Qur'an (i.e., the holy book of Islam) is "Read, in the name of your Lord" (The Qur'an

96:1). However, the Qur'an does not say *what* to read, just that reading is very important, especially for generating fresh thoughts about science [33].

The importance of and need for reading intensifies with increased information, a knowledge-based economy, technological developments, and progress after the recent industrial revolution. The need for reading proficiency increases when societies and cultures become more complex, discoveries and innovations increase, and modern technology advances. Reading is an urgent and necessary tool for people desiring social and economic progress.

The concept of reading has evolved over the years. At the beginning of the 20th century, reading constituted recognizing and knowing letters and words and correctly pronouncing them. During the next decade, scientific researchers began to study reading, with the most important contributions coming from Edward Thorndike (1874–1949), a pioneer in educational psychology. He concluded that reading is not just a mechanical process limited to mere recognition and articulation but rather a complex process that requires understanding, linking, and drawing conclusions. The importance of criticism, another element of reading, arose later [34].

During the third decade of the 20th century, reading expanded to include the benefits of the material being read in solving problems [34]. The concept of reading was further expanded to include three dimensions of the linguistic experiences that readers encounter: (1) the sensory dimension, which depends on the reader's background and previous experiences; (2) the emotional dimension, which includes feelings and emotional reactions during the reading process; and (3) the cognitive dimension, which includes mental skills for thinking, reasoning, judging, interpreting, and drawing conclusions. Reading is thus conceptualized as a means of intellectual and emotional development [35].

The purpose of teaching reading is to help students comprehend (i.e., understand and internalize) what they have read; without comprehension, people experience incomplete reading. This cannot be called reading in light of the concept of modern education. Comprehension is the main pillar of reading, and this relationship is made more challenging because reading and comprehension are two complex skills [36]. Without the ability to read, information cannot be acquired, let alone comprehended. Without comprehension, external information cannot be internalized as knowledge. If students cannot say "I get it," they cannot use it in their lives. This is *incomplete* reading [36].

Reading also helps students engage in dialogue with the text and ask questions to reach conclusions. This type of reading requires a high level of thinking [37]. Such reading helps individuals expand their perceptions and leads them to wider horizons. It helps them recreate and solve problems by making them think about what they are reading, compare different points of view, justify what needs interpretation, criticize the material being read, and judge the material at hand [35].

Reading is considered a common factor between language and science [38]. As with all other school subjects, learning science depends on reading and comprehension, especially in the elementary stage, where students learn the rudimentary elements of both subjects: language (including how to read) and science. To attain high levels of achievement in science, students must be able to read textbooks and process the questions asked in tests. Thus, reading is an important scientific activity. Students must be able to analyze scientific information and read it critically. After leaving school, they must be able to discern and judge scientific information in various media, especially written ones, such as newspapers, magazines, journals, books, and internet articles [39].

Students' scientific reading capability is considered a key factor affecting their learning and understanding of various scientific concepts. The importance of scientific reading is why teachers must take it into account and include it in their instructional planning and preparations to guide the educational process [40]. They concluded that science teachers did not generally show any trend (positive or negative) toward the implementation of reading in teaching science. They recommended paying attention to using reading in

science teaching, including scientific reading as a goal of science curricula, and following it up to see if teachers use reading when teaching science.

Teachers can employ reading in science education as an active learning method to achieve numerous educational goals: developing writing skills, dialogue, scientific interest, and an aptitude for science. As an active learning strategy, reading plays a positive role in the learning process because it helps students to learn science better. The more science they learn by reading, the more inclined they are to develop a habit of reading scientific texts in and beyond school.

*2.2. Reading Tendencies*

Generally, a tendency (i.e., an inclination) toward reading reflects learners' interest in reading different (a variety of) texts and materials. This tendency and interest foster a real desire to practice reading regardless of what is being read. This makes learners want to both participate in activities related to reading and feel comfortable with reading whether in general or in a specific subject such as science. Unsurprisingly, the concept of tendencies is considered complex, comprising several levels: curiosity, interest, and emotional attachment to the topic. However, a tendency toward reading leads a person to practice reading with intent and satisfaction regardless of what is being read [41].

Tendencies toward reading are predictably linked to the goals of teaching reading and helping students read, as tendencies guide teachers in choosing appropriate materials when planning effective programs to teach reading. Regardless of the grade level, teaching reading mainly aims to help learners find lasting pleasure and satisfaction in reading and be inclined to use what they learn in the real world [42]. Al-Anzi [43] believed that low levels of tendencies toward reading can be attributed to teaching strategies that do not support the development of reading comprehension. Unfortunately, students may be reluctant to read and practice reading. This aversion may hamper scientific learning.

Reducing reluctance and resistance to reading is thus imperative in science learning. Dawood [44] found that free reading (i.e., reading a book of choice at one's own pace) helped develop reading comprehension and some 21st century skills. He recommended the use of free reading to develop these skills. Al-Swaify [45] reported on the development of primary textbooks for 21st century skills with the intent of developing reading enlightenment skills and tendencies toward reading. Al-Maamouri [46] reported that using the 'Anthony model of guided reading' with third-year intermediate physics students made a difference in the achievement of scientific curiosity.

*2.3. Research Questions and Objectives*

Meeting the imperative of encouraging students to develop early on the habit of reading scientific texts will be challenging in the KSA, given that the results of the 2018 Programme for International Student Assessment (PISA) showed that 52% of participating Saudi students did not achieve the baseline for proficiency in reading to live in society [47]. This was the first time that the KSA had participated in PISA (alongside 79 other countries). Considering that reading is a key aspect of communication [7], it is important that Saudi educators develop learners' habits of reading scientific texts.

Accordingly, research must be conducted to identify Saudi science teachers' beliefs about the extent to which students have acquired these reading habits and students' thoughts on their own science reading habits. This study was guided by two research questions and related research objectives. The latter are tasks undertaken to collect data to answer the research questions [48]:

Research Question 1. What are Saudi female primary science teachers' beliefs about their female students' scientific reading habits?

Research Objective 1: Survey Saudi female primary science teachers using a researcher-developed instrument comprising belief-related questions.

Research Question 2. What effect does intentionally reading scientific texts have on developing elementary students' reading habits as perceived by students themselves?

Research Objective 2: Survey Saudi female primary science students using a researcher-developed and validated Likert-scale instrument.

## 3. Method

A descriptive research design was employed to develop a detailed description of a specific phenomenon or problem [48,49]. Originally, the researchers employed a mixed-method, repeated cross-sectional design (i.e., data collected once from different samples) combining (1) qualitative data (open-ended teacher questionnaire) and (2) quantitative student survey data. When the teachers' written responses were deemed too inadequate to generate trustworthy qualitative findings, the mixed-methods approach was set aside and teachers' data were treated as *quantitative* results.

### 3.1. Sample Frame

Study respondents were from two offices (Dammam and Al Khobar) in one Saudi Eastern Province school board that was geographically divided into 10 offices. Of $n = 234$ Saudi female primary school teachers, $n = 55$ agreed to participate in the study. The final student sample comprised two groups of female primary students (grades four and five). The first group (purposive sampling, $n = 41$) was from a private school (taught by three science teachers) in the school board office. They engaged in five 45-min classes per week, which included intentionally reading scientific texts. The second group (random sampling, $n = 191$) included public school students in the same school board but from two offices. They met three times a week for 45-min periods and did not intentionally engage in scientific reading, although some may have experienced this approach. Primary education in Saudi Arabia is currently segregated; it is a norm for female teachers to teach female students.

Female elementary school students were chosen because of their accessibility to the researchers. Female teachers were of scientific interest because they constitute more than half (52%) of all public school teachers in Saudi Arabia [50], especially primary school, where they constituted 100% in 2015 [51]. Their role in instilling science into Saudi citizens' collective psyche is profound. Islam places a high value on reading [33], yet nominal research exists in Saudi Arabia around the role of students' tendencies to read in general (reading habits) relative to their learning science. Additionally, female teachers were of interest because, until very, very recently, only women were allowed to teach girl students in Saudi primary school [52].

### 3.2. Data Collection Instruments

#### 3.2.1. Teachers' Instrument

To collect data to answer Research Question 1, the researchers developed an instrument comprising 13 questions about teachers' beliefs about the extent to which their students possessed the skills required to read scientific texts (see Table 1). The researchers envisioned those skills as comprising the 13 verbs used to frame the 13 questions: participating, posing questions, expressing interest, paying attention, discussing, acquiring, and so on. Teachers were invited to write out their answers to this open questionnaire, which dealt with students' acceptance of, enthusiasm with, happiness with, and eagerness for reading scientific texts. Teachers were also queried about the extent to which students engaged in reading scientific texts, both during and outside of class.

#### 3.2.2. Students' Instrument

To collect data for the second research question, the researchers developed an instrument to solicit information from students about their reading habits of scientific texts. An original, 26-item instrument with clear and plain language was developed so that it fit the verbal experience of primary school students. This version was pilot tested, and all items were retained because of a Cronbach's $\alpha$ coefficient of 0.972, with all items presenting a high stability coefficient (ranging from 0.970 to 0.971; see Table 2). In addition, all items achieved an acceptable value for internal validity (higher than 0.300) with coefficients ranging from

0.493 to 0.891. This instrument employed a 5-point Likert scale roughly translating to very large (5), large (4), average (moderate; 3), small (2), and negligible (1).

**Table 1.** Thirteen belief questions posed to Saudi female primary science teachers regarding students' possession of skills required to read scientific texts.

| | The Belief Questions |
|---|---|
| 1 | How enthusiastic are students to participate in reading scientific texts? |
| 2 | Do pupils seem happy when they participate in reading scientific texts? |
| 3 | Do pupils volunteer to read aloud in the science class in front of their classmates? |
| 4 | To what extent do pupils participate in classroom discussions around the topics they are reading? |
| 5 | To what extent do pupils pay attention to their classmates' reading of scientific texts? |
| 6 | Do pupils discuss the scientific topics they have read with you? |
| 7 | Do pupils ask you about the names of non-prescribed science books to read? |
| 8 | Do pupils acquire extensive information in the field of science? |
| 9 | Do pupils show an interest in electronic references and sources related to scientific topics? |
| 10 | To what extent do pupils know the names of scientific books and encyclopedias? |
| 11 | Do pupils read scientific texts eagerly? |
| 12 | To what extent do pupils show a clear understanding of the scientific texts they have read? |
| 13 | Do pupils ask questions in science class that express an eagerness to learn and understand scientific concepts? |

**Table 2.** Twenty-six items in the Saudi primary science students' survey instrument about their reading habits of scientific texts.

| | Items in Survey Instrument |
|---|---|
| 1 | How interested are you in reading scientific topics in general? |
| 2 | How much do you like to read magazines, websites, and scientific books? |
| 3 | How interested are you in buying scientific books and magazines? |
| 4 | To what extent are you keen to borrow scientific books and stories? |
| 5 | How much time do you want to set aside to read books and scholarly journals in the school library? |
| 6 | To what extent do you want a classroom library that includes science books and references? |
| 7 | To what extent do you find pleasure in reading about modern scientific inventions and discoveries? |
| 8 | How much do you love to read about science and scientists and their lives? |
| 9 | How much do you love to read about modern scientific inventions and discoveries? |
| 10 | How interested are you in reading topics dealing with space science? |
| 11 | How much do you enjoy reading in your free time on various scientific topics? |
| 12 | How important is it to you to collect photos and scientific drawings and read around them? |
| 13 | How often do you participate in writing for the scientific page in the school bulletin? |
| 14 | How interested are you in science stories told by the teacher or written in/on newspapers, magazines, and websites? |
| 15 | How much do you like to talk about your readings about scientific discoveries? |
| 16 | How much do you benefit from reading scientific texts? |
| 17 | To what extent do you want the science teacher to allocate time to present and discuss the scientific stories you are reading? |
| 18 | How much do you want the science teacher to ask you to summarize and display what you have read about scientific topics? |
| 19 | To what extent do you want to have free reading activities allocated as part of the science course? |
| 20 | How well do you feel that reading science helps you know the world around you? |
| 21 | How much do you want the school radio to present science subjects? |
| 22 | To what extent do you feel that reading in scientific fields enhances your scientific information? |
| 23 | To what extent do you feel that reading in scientific fields deserves great attention from you? |
| 24 | To what extent do you feel that reading in scientific fields is a form of scientific excellence? |
| 25 | To what extent do you feel that reading is an essential means to inform you of scientific news? |
| 26 | To what extent do you monitor or view websites related to scientific subjects? |

The 26 items were organized along two subdimensions: students' habits of reading scientific texts (1) while at school, in class, and in the library (*academic science*) and (2) in their daily life outside of school (*life science*). The latter concerned their general interest in procuring and reading scientific texts as well as learning about scientists, discoveries, and inventions that shaped their world. This operationalization reflected both (1) the authors' observations that Saudi female primary school students rarely connect science with reading

and (2) Butler's [23] assertion that reading science texts in the classroom (academic science) can be a precursor to students reading science texts in their daily lives to make sense of their world (life science).

The 26 survey items and two subdimensions were an attempt to approach this phenomenon from the students' perspective. Table 3 examines which test items pertained to each subdimension. The instrument had a high degree of validity according to the *r* values. The stability of the two subdimensions with the psychometric indicators of the instrument and its reliability in field application were verified.

**Table 3.** Profile of instrument scale items by subdimension, instrument stability, and validity.

| Subdimension | Items Related to the Subdimension | Correlation Coefficient (*r* Value) | Coefficient ($\alpha$) per Dimension (Scale as a Whole: 0.972) |
|---|---|---|---|
| Life Science | 1–2–3–7–8–9–11–12–15–20–24–25–26 | 0.988 | 0.947 |
| Academic Science | 4–5–6–10–13–14–16–17–18–19–21–22–23 | 0.985 | 0.947 |

### 3.3. Data Collection

Data from teachers were collected in March 2020, during the COVID-19 global lockdown, using the 13-question instrument developed for the study (see Table 1). The researchers emailed the open-ended questionnaire to the school board's office, which, in turn, forwarded it to $N = 234$ female Saudi teachers. Participation was voluntary with $n = 55$ respondents emailing their answers to the second author. The 24% response rate is acceptable for an email survey, which tends to average 10% to 30%. The survey completion constituted consent [53–55].

Student data were also collected in March 2020. Regarding the procedure, after uploading the web-based survey (see Table 2) to *Google Forms*, the second author sent an email containing the URL to the school board, which forwarded it to school principals, who sent it to teachers, asking them to send it to students. Students' participation was voluntary, and survey completion constituted consent [54,55].

As a caveat, there was no way to determine which students had Internet access during the COVID-19 lockdown. Without such information, it was impossible to calculate the final response rate, which is normally 25–30% for a web-based survey [53,55]. That said, the completion rate after the data were cleaned (i.e., incomplete surveys removed; [56,57]) was 77%, which is very close to the expected rate of 74% for 26 questions (based on 85% for 30 questions). High completion rates indicated that the data used to answer the main research question were reliable [58,59].

### 3.4. Data Analysis

Teachers' qualitative data were converted to numbers [48] due to teachers' insufficient written responses, as previously mentioned, and then quantitatively analyzed using descriptive statistics (frequencies, percentages, means, standard deviation, Chi-square). Student survey data ($n = 232$) were also analyzed using descriptive statistics (frequencies, percentages, means, standard deviations, and *t*-tests).

## 4. Results

### 4.1. Teachers' Beliefs about Students' Scientific Reading Habits

The first research question focused on Saudi female primary school science teachers' beliefs regarding the scientific reading habits of their female students. Teachers responded in writing to 13 questions (see Table 1). As explained, because the data were not adequate to ensure trustworthy qualitative findings, they were converted into numbers, analyzed using descriptive statistics, and reported herein as quantitative results.

Four of the teachers' 13 beliefs (see Table 1) are reported in this paper. They included (1) the nature of the questions students asked in class and the extent to which these expressed their eagerness to learn and understand scientific concepts, (2) students' habit

of reading scientific texts, (3) students' desire to participate in discussions about scientific topics they had read about, and (4) students' ability to comprehend and understand what they had read. Overall, teacher respondents believed that students had a high degree of eagerness and passion to learn science, were always passionate and eager to read scientific texts, had a high desire to participate in class discussions on scientific readings, and presented a moderate understanding of what they had read. All results were statistically significant.

First, based on the nature of the questions asked by their students, more than half of the teachers (67.27%) believed that students had a high degree of eagerness to learn and understand science. Combined with a *very high degree* of eagerness, this result increased to 81.9%. Very few teachers (18.1%) reported feeling otherwise (low to no eagerness). This difference was statistically significant (see Table 4).

**Table 4.** Teachers' beliefs about the extent to which questions students asked in class expressed their eagerness to learn and understand science.

| Very High Degree of Eagerness | | High Degree of Eagerness | | Medium Degree of Eagerness | | Low Degree of Eagerness | | No Eagerness | | Arithmetic Mean | Standard Deviation | Chi | Degrees of Freedom | Significance Level |
|---|---|---|---|---|---|---|---|---|---|---|---|---|---|---|
| *n* | % | *n* | % | *n* | % | *n* | % | *n* | % | | | | | |
| 8 | 14.55 | 37 | 67.27 | 4 | 7.27 | 3 | 5.45 | 3 | 5.46 | 2.20 | 0.951 | 78.364 | 4 | 0.000 |

Second, teachers tended to believe that half of the students (53%) were *always* passionate about and eager to read scientific texts with less than one-third (29%) reporting that this is true only *sometimes*. Approximately one-fifth (18%) believed that students were not passionate or eager to read scientific texts. This difference was statistically significant (see Table 5).

**Table 5.** Teachers' beliefs about extent to which students were passionate about and eager to read scientific texts.

| Always Passionate and Eager | | Sometimes Passionate and Eager | | Never Passionate or Eager | | Arithmetic Mean | Standard Deviation | Chi | Degrees of Freedom | Significance Level |
|---|---|---|---|---|---|---|---|---|---|---|
| *n* | | *n* | | *n* | | | | | | |
| 29 | 52.73% | 16 | 29.09% | 10 | 18.18% | 1.65 | 0.775 | 10.291 | 2 | 0.006 |

Third, teachers believed that students had a high desire (52.73%) to participate in classroom discussions about scientific readings. This was closely followed by respondents who judged students' desire to be moderate (45.5%). The difference was statistically significant (see Table 6).

**Table 6.** Teachers' beliefs of students' desire to participate in classroom discussions about scientific topics.

| High Desire to Participate | | Medium Desire to Participate | | Low Desire to Participate | | Arithmetic Mean | Standard Deviation | Chi | Degrees of Freedom | Significance Level |
|---|---|---|---|---|---|---|---|---|---|---|
| *n* | | *n* | | *n* | | | | | | |
| 29 | 52.73% | 25 | 45.45% | 1 | 1.81% | 1.49 | 0.540 | 25.018 | 2 | 0.000 |

Finally, half (50.9%) of the teachers believed that students could moderately understand what they had read. Approximately one-third (30.9%) rated this understanding as high or very high. Less than one-fifth (18.2%) rated it as low or very low. This difference was statistically significant (see Table 7).

**Table 7.** Teachers' beliefs about students' ability to understand scientific texts they are reading.

| Very High Understanding | | High Understanding | | Medium Understanding | | Low Understanding | | Very Low understanding | | Arithmetic Mean | Standard Deviation | Chi | Degrees of Freedom | Significance Level |
|---|---|---|---|---|---|---|---|---|---|---|---|---|---|---|
| *n* 8 | 14.55% | *n* 9 | 16.36% | *n* 28 | 50.91% | *n* 7 | 12.73% | *n* 3 | 5.45% | 2.78 | 1.031 | 34.727 | 4 | 0.000 |

### 4.2. Students' Survey Results

The second research question dealt with the effect of intentionally reading scientific texts on developing elementary students' self-perceived reading habits. As expected, there were differences depending on the type of science education received. Students who did *not* intentionally read science (public school students) scored lower than those who did so intentionally (private school students), mean = 3.48 and 3.80, respectively (see Table 8). The *t*-test for the individual groups revealed no statistically significant differences between the overall results and the two subdimensions of science reading (academic science and life science; see Table 8).

**Table 8.** Students' overall results (mean and *t*-test), type of school and science education, and subdimension of science reading.

| Overall | Public School (Not Intentionally Reading Science) | | Private School (Intentionally Reading Science) |
|---|---|---|---|
| Mean = 3.64 (Moderate) | Mean = 3.48 (Moderate) | | Mean = 3.80 (Moderate tending to large) |
| | *t*-test | Degrees of Freedom | Significance |
| Overall Science Reading | 1.00 | 230 | 0.318 |
| Science Reading for Academics | 0.975 | 230 | 0.330 |
| Science Reading for Life | 1.33 | 230 | 0.184 |

### 4.2.1. Private Schools Intentionally Taught Reading Science

Table 9 shows the results for the 41 private school students who were intentionally taught science education using the strategy of reading science (see Table 2 for the description of each item). None of these results was statistically significant (Table 8).

**Table 9.** Private school students intentionally reading science/reading for life versus academics (*n* = 41, mean = 3.80).

| Reading Science in Daily Life | | | | | | Reading Science in School (Academic) | | | | | |
|---|---|---|---|---|---|---|---|---|---|---|---|
| Item | Likert Category | % | *n* | Overall Mean | SD | Item | Likert Category | % | *n* | Overall Mean | SD |
| 1 | Average Large/Very large | 41.2 53.7 | 17 22 | 3.73 | 0.949 | 4 | Average Large/Very large | 31.8 41.5 | 13 17 | 3.34 | 1.153 |
| 2 | Large/Very large | 58.5 | 24 | 3.71 | 1.167 | 5 | Large Large/Very large | 41.5 65.9 | 17 27 | 3.78 | 0.962 |
| 3 | Large/Very large | 51.2 | 21 | 3.39 | 1.070 | 6 | Very Large Large/Very large | 41.5 73.1 | 17 30 | 4.05 | 0.999 |
| 7 | Large/Very large | 70.7 | 29 | 4.15 | 0.937 | 10 | Very Large Large/Very large | 39 68.3 | 16 28 | 3.95 | 1.048 |
| 8 | Large/Very large | 65.9 | 27 | 3.88 | 1.005 | 13 | Average Large/Very large | 31.8 41.5 | 13 17 | 3.24 | 1.300 |
| 9 | Very large | 51.2 | 21 | 4.10 | 1.068 | 14 | Large/Very large | 61 | 25 | 3.71 | 1.078 |
| 11 | Large/Very large | 53.7 | 22 | 3.46 | 1.075 | 16 | Very Large Large/Very large | 41.5 68.3 | 17 28 | 3.98 | 1.060 |
| 12 | Large/Very large | 56.1 | 23 | 3.61 | 1.181 | 17 | Very Large Large/Very large | 39 58.5 | 16 24 | 3.76 | 1.261 |
| 15 | Large/Very large | 56.1 | 23 | 3.63 | 1.090 | 18 | Large/Very large | 51.2 | 21 | 3.51 | 1.267 |
| 20 | Large/Very large | 85.4 | 35 | 4.22 | 0.881 | 19 | Large/Very large | 63.4 | 26 | 3.88 | 1.005 |

**Table 9.** *Cont.*

| | Reading Science in Daily Life | | | | | | Reading Science in School (Academic) | | | | |
|---|---|---|---|---|---|---|---|---|---|---|---|
| Item | Likert Category | % | *n* | Overall Mean | SD | Item | Likert Category | % | *n* | Overall Mean | SD |
| 24 | Very large<br>Large/Very large | 43.9<br>78.1 | 18<br>32 | 4.17 | 0.892 | 21 | Very Large<br>Large/Very large | 39<br>58.5 | 16<br>24 | 3.83 | 1.138 |
| 25 | Large<br>Large/Very large | 41.5<br>75.6 | 17<br>31 | 4.00 | 0.922 | 22 | Very Large<br>Large/Very large | 44<br>75.6 | 18<br>31 | 4.10 | 0.995 |
| 26 | Large/Very large | 53.4 | 22 | 3.59 | 1.204 | 23 | Large<br>Large/Very large | 41.5<br>68.3 | 17<br>28 | 3.83 | 0.998 |
| Total | *Average/Moderate*<br>*(tending to Large)* | | | 3.82 | | | *Average/Moderate*<br>*(tending to Large)* | | | 3.77 | |

Virtually all items scored as large/very large (combined). Six items had percentage scores that were 70% or higher. Students intentionally taught using science reading said this helped them better understand their world (85.4%), it is a form of scientific excellence (78.1%), it is essential to gain scientific news (75.6%), and it enhances their scientific information (75.6%). They wanted a science library in their classroom (73.1%), and they found it pleasurable to read about scientific inventions and discoveries (70.7%; Table 9).

The remaining items (*n* = 20) scored as large/very large combined but with lower percentage scores ranging from 50–69%. Most of these (above 65%, *n* = 5) pertained to students agreeing that reading science deserves attention and is beneficial. They were interested in reading about space science and scientists and their lives, and they wanted time to read scientific books and journals in their school library. Although still scoring as large/very large combined, two items had percentage scores below 50%: borrowing science books (item 4) and writing for the science section of their school bulletin (item 13; see Table 9).

### 4.2.2. Public Schools Not Intentionally Teaching Reading Science

Compared to private school students, public school students in this study had a lower mean score (3.80 and 3.48, respectively; *moderate*; see Table 8) and scored highest on different items. Similar to private school students, nearly all frequency scores were large/very large combined, but the percentages were much lower (see Table 10).

**Table 10.** Public school not intentionally reading science/reading science for life versus academics (*n* = 191, mean = 3.48).

| | Reading Science in Daily Life | | | | | | Reading Science in School (Academic) | | | | |
|---|---|---|---|---|---|---|---|---|---|---|---|
| Item | Likert Category | % | *n* | Overall Mean | SD | Item | Likert Category | % | *n* | Overall Mean | SD |
| 1 | Average<br>Large/Very large | 41.2<br>53.7 | 17<br>22 | 3.73 | 0.949 | 4 | Average<br>Large/Very large | 31.8<br>41.5 | 13<br>17 | 3.34 | 1.153 |
| 2 | Large/Very large | 58.5 | 24 | 3.71 | 1.167 | 5 | Large<br>Large/Very large | 41.5<br>65.9 | 17<br>27 | 3.78 | 0.962 |
| 3 | Large/Very large | 51.2 | 21 | 3.39 | 1.070 | 6 | Very Large<br>Large/Very large | 41.5<br>73.1 | 17<br>30 | 4.05 | 0.999 |
| 7 | Large/Very large | 70.7 | 29 | 4.15 | 0.937 | 10 | Very Large<br>Large/Very large | 39<br>68.3 | 16<br>28 | 3.95 | 1.048 |
| 8 | Large/Very large | 65.9 | 27 | 3.88 | 1.005 | 13 | Average<br>Large/Very large | 31.8<br>41.5 | 13<br>17 | 3.24 | 1.300 |
| 9 | Very large | 51.2 | 21 | 4.10 | 1.068 | 14 | Large/Very large | 61 | 25 | 3.71 | 1.078 |
| 11 | Large/Very large | 53.7 | 22 | 3.46 | 1.075 | 16 | Very Large<br>Large/Very large | 41.5<br>68.3 | 17<br>28 | 3.98 | 1.060 |
| 12 | Large/Very large | 56.1 | 23 | 3.61 | 1.181 | 17 | Very Large<br>Large/Very large | 39<br>58.5 | 16<br>24 | 3.76 | 1.261 |
| 15 | Large/Very large | 56.1 | 23 | 3.63 | 1.090 | 18 | Large/Very large | 51.2 | 21 | 3.51 | 1.267 |

**Table 10.** *Cont.*

| | Reading Science in Daily Life | | | | | | Reading Science in School (Academic) | | | | |
|---|---|---|---|---|---|---|---|---|---|---|---|
| **Item** | **Likert Category** | **%** | **n** | **Overall Mean** | **SD** | **Item** | **Likert Category** | **%** | **n** | **Overall Mean** | **SD** |
| 20 | Large/Very large | 85.4 | 35 | 4.22 | 0.881 | 19 | Large/Very large | 63.4 | 26 | 3.88 | 1.005 |
| 24 | Very large<br>Large/Very large | 43.9<br>78.1 | 18<br>32 | 4.17 | 0.892 | 21 | Very Large<br>Large/Very large | 39<br>58.5 | 16<br>24 | 3.83 | 1.138 |
| 25 | Large<br>Large/Very large | 41.5<br>75.6 | 17<br>31 | 4.00 | 0.922 | 22 | Very Large<br>Large/very large | 44<br>75.6 | 18<br>31 | 4.10 | 0.995 |
| 26 | Large/Very large | 53.4 | 22 | 3.59 | 1.204 | 23 | Large<br>Large/Very large | 41.5<br>68.3 | 17<br>28 | 3.83 | 0.998 |
| Total | *Average/Moderate<br>(tending to Large)* | | | 3.82 | | | *Average/Moderate<br>(tending to Large)* | | | 3.77 | |

Two items had average scores (i.e., the combined large/very large scores were not higher) and pertained to moderate interests in reading science topics (44.3%, item 1) and borrowing scientific books (36.9%, item 4). Otherwise, virtually all items scored as large/very large (combined). Only two items had percentage scores of 70% or higher. Reading science was moderately essential for gaining scientific news (71.7%, item 25), and students wanted a science library in their classroom (71.3%, item 6; see Table 10).

The remaining (84%) item percentage scores ranged from 32.5% to 69.5% with nine items (one-third) scoring between 60% and 69% (means ranging from 2.95 and 4.01 and averaging 3.48). Reading science moderately helped public school students to better know their world and enhance their scientific information, in that order. They moderately felt that science was deserving of their attention, and they wanted to read both science stories identified by the teacher and those found in the media and on the Internet. They were also moderately interested in reading about modern scientific inventions and discoveries (see Tables 2 and 10).

*4.3. Science Reading for Academics versus for Life*

Regarding the overall mean scores, respondents were marginally more inclined to read science for academics (school; mean = 3.69) than science for life (3.59); the average/moderate scores were quite close. Scores for reading science for life were higher among private than public school students: 3.82 and 3.36, respectively. The same pattern held for reading science for academics: 3.71 and 3.60, respectively (see Tables 9 and 10).

Furthermore, students who learned without intentionally reading science tended to prefer reading science for academics (mean = 3.60) rather than science for life (mean = 3.36). Students intentionally reading science presented the same profile (3.77 and 3.82, respectively) but with higher scores and a smaller differential gap (see Tables 9 and 10).

4.3.1. Science Reading for Academics

Regarding reading science in school, students who intentionally read science scored highest on items 22 and 6. Respectively, they strongly agreed (large) that reading science enhanced their scientific information (mean = 4.10) and they wanted a science library in their classroom (mean = 4.05). Students who did not intentionally read science also scored highest on items 6 and 17 (moderate tending to large). They also wanted a science library in their classroom (mean = 3.99), and they wanted the teacher to make time to discuss what they had read (mean = 3.85).

Both public and private school students scored lowest on the same two items: being moderately interested in borrowing science books to read (item 4, mean = 3.07 and 3.34, respectively) and contributing scientific articles to the school bulletin (item 13, mean = 2.95 and 3.24, respectively).

### 4.3.2. Science Reading for Life

Students who intentionally read science scored the highest on items 20, 24, and 25. They strongly (large) agreed that reading science helped them better understand their world (mean = 4.22); they viewed it as a form of scientific excellence (mean = 4.17), and it was essential to acquiring scientific news (mean = 4.00). Students who did not intentionally read science scored highest on two similar items (20, 25) and item 7 (but lower than private school students). They strongly (large) agreed that reading science helped them better understand their world (mean = 3.98) and was essential to informing them about scientific news (mean = 4.01). They were also interested in reading about modern scientific inventions and discoveries (mean = 3.95).

Both public and private school students scored the lowest on the same two items: being moderately interested in buying science magazines and books to read (item 3, mean = 3.10 and 3.38, respectively) and enjoying reading science topics in their free time (item 11, mean = 3.38 and 3.46, respectively).

### 5. Discussion

The research problem underpinning this study was the role of reading science in achieving quality science education that supports 21st century skills. Regarding the first research question, Saudi female science teachers held fairly strong beliefs that primary female students had a (1) high degree of eagerness and passion to learn science and read scientific texts, (2) high desire to participate in class discussions of scientific readings, and (3) moderate understanding of what they had read. Although not directly asked about 21st century skills, these results indicate that teachers were aware of these skills. This is in line with Al-Harbi and Al-Jabr's (2016) conclusions. Future research should explore this topic further by focusing on male primary science teachers and male students and on the very recent phenomenon of women teaching primary male students [52].

Our results are encouraging as elementary science teachers in general tend to agree that students must be able to read science because they learn science better this way [21] and, by association, 21st century skills. Furthermore, all Saudi teachers and science students are expected to develop 21st century skills [12,13], which is most likely to happen if teachers teach science by employing science reading [21].

Regarding the second research question, students who were intentionally taught science reading scored higher (mean = 3.80) in their reading habits than public school students (mean = 3.48). This result mirrors the literature, wherein scholars proposed that students who learn science by reading science will have stronger reading tendencies [41,43]. Both results were moderate with the mean for private school students tending to *large* (3.80; see Table 8).

However, why is the private school mean score not higher, and what does it mean that the results were not statistically significant (see Table 8)? The latter conventionally means that results happened by chance and have little to no value when addressing the research question [60]. However, Drotar [61] asserted that, although not statistically significant, the results can still be *significant* if they address other important questions.

With this caveat in mind, consider that learning science depends on reading and comprehension, especially in elementary schools where students learn the basics [62]. Saudi science teachers, however, tend not to use a student-centered approach that includes reading science [16,17], which is purportedly the best way to instill 21st century skills [7,29,31]. One would assume that intentionally using a student-centered pedagogy (science reading) would instill a higher tendency to read than indicated by the moderate mean score. Moreover, because they were more likely to be taught by someone not inclined to have them read science, public school students should have scored lower than they did (average mean = 3.48). These results may partially be explained by the Saudi context, wherein Islam places a high value on reading regardless of the pedagogical approach employed in science education [33].

Ambo and Al-Arimi [40] reported that science teachers do not generally show any trend (positive or negative) toward the use of reading in teaching science. The private-school teachers in our study were purposefully chosen because the researchers had determined that they used reading to teach science. However, the randomness of the public school sampling procedure prevented the researchers from knowing these teachers' inclination to use reading to teach science. The lower-than-expected result for private school students and the higher-than-expected result for public school students (see Table 8) strongly suggested that Saudi teachers presented a mix of preferences for using reading to teach science [40]. Future researchers should consider this variable when designing studies.

Osborne [22] noted that, even when elementary students received a good start on reading, they often lacked skills for reading science, which is a very complex skill set. Saudi students are already accustomed to reading in their social science and language courses, and Islam expects them to read outside school [33]. It is possible that these factors are so strong that the reading habit is transferred to the science class. This is a topic for future research.

The fact that respondents who were intentionally taught using scientific readings scored only moderately (mean = 3.80) on reading habits suggested that all Saudi elementary science teachers must shift pedagogical styles, so they can ensure that students are inclined to read science. Reading is the backbone of communication and one of the 4 Cs [7]. Shifting pedagogical styles ensures that students avoid incomplete science readings, which compromises their ability to transform science information into scientific knowledge that can be used in school and life [36].

Furthermore, scores at or below the Trends in International Mathematics and Science Study's (TIMSS) low benchmark (400) and PISA [24,47] indicate that Saudi primary students are not learning science. Moderate science reading habits herein appear to explain Saudi grade four students' low TIMSS and PISA science scores. Reading science improves science learning [43]. This well-accepted relationship supports the recommendation that Saudi teachers should be reoriented to science reading as a pivotal learning activity in science education [40]. They should be encouraged to integrate 21st century skills into the science curriculum [9].

Reading science texts helps students build an in-depth understanding of science concepts and principles [21], which half of the teachers in this study thought was moderately in place. Based on the scores for item 22 (i.e., reading science enhances one's scientific information), students also rated this ability as moderate tending to large (overall mean = 3.95). This middle-of-the-road result suggests a rich opportunity to use science reading to teach science thereby improving reading and comprehension, which is one of the 4Cs [7].

Most (81.9%) teacher respondents believed that students had a high or very high degree of eagerness and passion for reading science, but students themselves seemed more ambivalent, scoring as moderate (overall mean = 3.64, see Table 8). As noted, very few Saudi science teachers teach science using science reading [16,17]; however, teachers in this study believed that students were passionate about this learning activity while students themselves expressed a moderate interest.

To explain this apparent disconnect, future research could explore the dynamics of teachers psychologically projecting their own passion for science onto students; if they value science reading, their students will, too. Ironically, operating on this assumption may prevent them from intentionally using science reading as a learning strategy—a missed pedagogical opportunity to eventually raise mean scores to higher levels. Another research recommendation is, thus, to ensure that both teachers and students are from the same classes, wherein students' answers would reflect their actual teachers' practices and vice versa.

Any lack of interest in or resistance to science reading on the students' part must be addressed because resistance to reading science can hamper science learning [43], and resistance to reading in general hinders the acquisition of 21st century skills that are learned so effectively in science classes [9]. Free reading is one way to address this tendency [44].

Saudi primary students were moderately (mean = 3.42, item 11) interested in reading science in their free time (a form of free reading). This result presents an opportunity to foster this type of learning activity or to conduct research around the veracity of this recommendation in the Saudi context.

Finally, Butler [23] proposed that academic science (reading science in school) can be a precursor to students reading science texts in their daily lives to make sense of their world (life science). The results (Tables 9 and 10) marginally supported this proposed directional relationship with students scoring slightly higher on reading science for academics (mean = 3.68) than science for life (mean = 3.59). They also scored high (large) on item 20 (mean = 4.10), indicating their agreement that reading science helped them to understand the world around them.

## 6. Conclusions

It is imperative that students develop the early habit of reading scientific texts, so they can begin to acquire 21st century skills [21]. It is important to teach science using reading and instill the same in students while respecting this learning strategy. The results suggested that although Saudi female primary teachers fully believed that students generally embraced science reading, the students themselves were more ambivalent. Furthermore, the finding that those not intentionally taught to read science scored as high as those intentionally taught to read science may compel scholars to explore this phenomenon in the Saudi primary school context. That is, what is it about the Saudi educational context that such a finding was manifested? Is it the role of religion (Islam), teachers' pedagogical styles, national commitment to enriching human capital vis-à-vis Vision 2030, or some other collection/combination of factors?

Saudi Arabia is joining the international trend of reforming education so that all students can learn 21st century skills. Reading science to increase reading tendencies could be a key component of this strategy. The nation will benefit from citizens who are avid and critical readers with a deep respect for science learning and reading skills and citizens who can communicate and collaborate with others.

## 7. Contributions to the Literature

This study is the first attempt at addressing Saudi female primary school teachers' beliefs about Saudi students' reading habits in the science classroom. It ties results to the imperative of learning 21st century skills: critical thinking, creativity, collaboration, communication. Teachers' positive beliefs were belied by students' ambivalence. More work needs to be done.

**Author Contributions:** Conceptualization, A.K.H.A. and W.H.A.; methodology, A.K.H.A.; validation, W.H.A.; formal analysis, A.K.H.A. and W.H.A.; investigation, A.K.H.A.; resources, A.K.H.A.; data curation, A.K.H.A.; writing—original draft preparation, A.K.H.A.; writing—review and editing, A.K.H.A. and W.H.A.; visualization, A.K.H.A. and W.H.A.; supervision, A.K.H.A. and W.H.A.; project administration, A.K.H.A. and W.H.A.; funding acquisition, W.H.A. All authors have read and agreed to the published version of the manuscript.

**Funding:** This research project was funded by the Deanship of Scientific Research, Princess Nourah bint Abdulrahman University, through the Program of Research Project Funding After Publication, grant No (PRFA-P-42-2).

**Informed Consent Statement:** Informed consent was obtained from all subjects involved in the study.

**Data Availability Statement:** The data that support the results of this study are available from the corresponding author, [AKAK], upon reasonable request.

**Conflicts of Interest:** The authors declare no conflict of interest.

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
