# Peer review of "Teaching 21st Century Skills in Saudi Arabia with Attention to Elementary Science Reading Habits"

_education, doi:10.3390/educsci12060392_

Round 1
Reviewer 1 Report
The authors of the article have well defined the problem, objectives, and research questions. The results are difficult to generalize due to the sample sizes and the fact that the study focuses on a very particular geographic area. However, I believe that the work is novel, may be of interest and opens an important line of research about science learning and science skills development in Saudi Arabia.
However, I think there are some aspects that should be clarified by the authors before their manuscript can be published:
- it is not clear to me why the sample consists of females and what is the scientific interest of restricting the sample to females. This should be explained by the authors.
- Have the authors made any comparisons with the perceptions of males? Perhaps this aspect could be discussed in the Discussion?
- Line 306 defines two subdimensions of the global scale. How have these subdimensions been defined? Has some kind of factor analysis been done or are they variables previously defined by the authors, from which the instrument has been designed?
- The sample of teachers is too small, in my opinion, to analyze the results by means of parametric tests. In my opinion, non-parametric tests should be applied.
- I suggest including a graph to clarify the results presented in the tables.
Author Response
Could you please see the attached PDF?

Reviewer 2 Report
Dear Authors:
Overall, the manuscript is clear, well written and structured. The literature review presented is up to date. Furthermore, the contribution is pertinent and of the interest to the theme.
Below are some (suggested) points that we would like commented on/improve by the authors:
- In section 3.2 there is a need that of clarifying the criteria for choosing these questions (table 1);
-Presenting graphical of the results improve the manuscript.
Rew
Author Response
Could you please see the attached PDF?

Round 2
Reviewer 1 Report
From my point of view, the authors have taken my comments into consideration and have answered all my questions. My doubts have been clarified and the changes that the authors have made in their manuscript have improved the quality of the article. The results are interesting and increase knowledge. The methods are correct. In addition, I believe that the results may be of interest to many researchers. My congratulations to the authors for their work.